# A Low-Cost Robotic Camera System for Accurate Collection of Structural Response

**Rolands Kromanis \***  **and Christopher Forbes**

School of Architecture, Design and the Built Environment, Nottingham Trent University, Nottingham, NG1 4FQ, UK

**\*** Correspondence: rolands.kromanis@ntu.ac.uk

**Abstract:** Vision-based technologies are becoming ubiquitous when considering sensing systems for measuring the response of structures. Availability of proprietary camera systems has opened up the scope for many bridge monitoring projects. Even though structural response can be measured at high accuracies when analyzing target motions, the main limitations to achieving even better results are camera costs and image resolution. Conventional camera systems capture either the entire structure or large/small part of it. This study introduces a low-cost robotic camera system (RCS) for accurate measurement collection of structural response. The RCS automatically captures images of parts of a structure under loading, therefore, (i) giving a higher pixel density than conventional cameras capturing the entire structure, thus allowing for greater measurement accuracy, and (ii) capturing multiple parts of the structure. The proposed camera system consists of a modified action camera with a zoom lens, a robotic mechanism for camera rotation, and open-source software which enables wireless communication. A data processing strategy, together with image processing techniques, is introduced and explained. A laboratory beam subjected to static loading serves to evaluate the performance of the RCS. The response of the beam is also monitored with contact sensors and calculated from images captured with a smartphone. The RCS provides accurate response measurements. Such camera systems could be employed for long-term bridge monitoring, in which strains are collected at strategic locations, and response time-histories are formed for further analysis.

**Keywords:** robotic camera; image processing; vision-based deformation monitoring; precision movement control; static measurement collection; non-contact sensor systems; photogrammetry

---

## 1. Introduction

Aging infrastructure needs a prudent and accurate assessment for the assurance of its components, such as bridges being fit for purpose and safe to use. Structural health monitoring (SHM) deals with measurement collection and interpretation, thus providing means of capturing response and dealing with challenges related to measurement interpretation and condition assessment [1]. The first challenge in SHM is the collection of reliable measurements. Developments in technologies have facilitated the evolution of sensors. Fiber optic sensors, wireless sensors, sensing sheets, and global positioning systems are just a few of the sensing technologies successfully employed in bridge monitoring [2–5]. The installation of contact sensors requires direct access to a structure, which may be disruptive and expensive, and involve working at height. Non-contact sensing systems such cameras and lasers have advantages over conventional contact sensor systems, especially when considering access to the structure and system installation as well as maintenance costs.

Robotic total stations collect accurate bridge response [6]. Typically, they have a displacement accuracy of 1 mm + 1.5 ppm for static measurements at a range of up to 1500 m. These systems need installation of a reflector on a bridge and can track movements of only one reflector. Image assisted

total stations have integrated cameras fitted to a telescope [7,8]. They can achieve 0.05–0.2 mm accuracy at a distance of 31 m in the laboratory environment and accurately identify frequencies of bridges in the filed trails [9]. Robotic stations cost between £20,000 and £95,000, making them an unattractive option, particularly when choosing a monitoring system for small to medium size bridges.

Structural response can be accurately measured from image frames (or videos), when analyzed with adequate proprietary or open-source algorithms, which are collected with low-cost cameras such as action cameras, smartphones, and camcorders [10–13]. Accurate measurements of multiple artificial or natural targets can be obtained [14]. Cameras with zoom lenses can measure small localized bridge displacements at accuracies similar to contact sensors [15–17]. An important factor affecting the measurement accuracy, besides camera stability, environmental effects such as rain and heat haze and image processing algorithms, is the number of pixels in the camera field of view. Usually, in bridge monitoring, only a small part of the structure is considered, thus providing very localized response measurements [18].

There is a need for very high-resolution images or camera systems capturing multiple closely zoomed parts of a structure to further improve measurement accuracy and capture response of the entire structure or parts of interest. Besides, this has to be achieved at a low cost to attract bridge owners' and inspectors' interest. Availability of open-source software and hardware has opened opportunities to create robotic systems for a range of computer vision applications such as autonomous real-time maneuvering of robots [19]. Highly accurate robotic camera systems have been successfully employed in laparoscopic surgeries [20]. However, these systems are cost-prohibitive and not suitable for far-range imaging applications such as bridge monitoring.

We propose to develop an open-source and low-cost robotic camera system capable of accurately and repeatedly capturing images of parts of a structure under monitoring. The primary purpose of the proposed camera system is to accurately capture slight changes in response such as vertical displacements and strains, which are difficult, if not impossible, to obtain for the entire structure using conventional vision-based systems. Additionally, close images of a structure may reveal cracks, which can be closer inspected either during visual inspections or using bridge inspection robots [21].

The performance of the proposed camera system is evaluated on a laboratory structure. The paper is structured as follows: Section 2 introduces the robotic camera system (hardware and software) and the data processing strategy, which includes data sorting, image processing, and response generation; in Section 3 the performance of the camera system is evaluated on a laboratory beam; Section 4 discusses the experimental findings, provides a vision of an enhanced three-axis robotic gimbal and gives an insight to future research; and Section 5 draws the main research findings.

## 2. Materials and Methods

A robotic camera system (RCS) is developed to collect accurate static and quasi-static structural response of an entire structure. A case of (a) static response is when a structure is loaded for a short period such as static calibrated load testing of a bridge with strategically positioned trucks [22]; (b) quasi-static response is when temperature loads force structures or their parts to expand or contract. A vision of an RCS application for the collection of bridge response for its condition assessment is given in Figure 1. The RCS is located at a suitable distance to the bridge, which depends on the camera lens, accessibility, and other factors. Close image frames of the entire bridge or selected parts of it are captured at regular intervals. Images are sent to a data storage unit, from which the data is processed. The structural response is the output of steps involved in the data processing phase. The response can then be analyzed for anomaly events and the structure's health. The robotic camera system and steps involved in the data processing phase are provided and discussed in the following subsections.

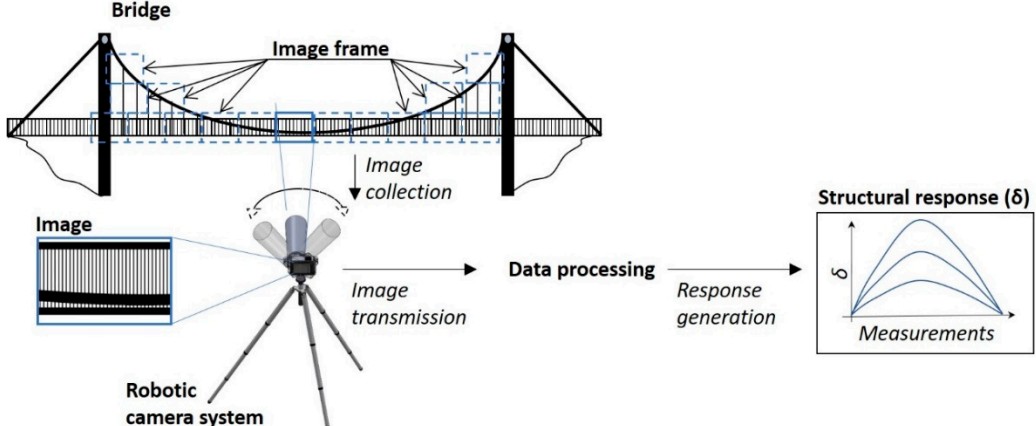

**Figure 1.** A robotic camera system for structural response collection.

## 2.1. Robotic Camera System

The proposed low-cost RCS, which combines a modified GoPro camera fitted with a long-range 1/2″ 25–135 mm F1.8 C-mount lens, a robotic mount, and software controls, is shown in Figure 2. The robotic mount uses a NEMA 17 stepper motor fitted to a 1:4 ratio gearbox providing enough torque and positional locking to rotate and hold the camera with the lens in the required position. The power is supplied via a USB cable connected to the mains or battery. The camera is positioned on the robotic mount via a lens holder arm keeping the camera sensor central to the rotational (vertical) axis. The main bracket of the robotic mount is designed to slide into a standard camera tripod where it is securely fixed.

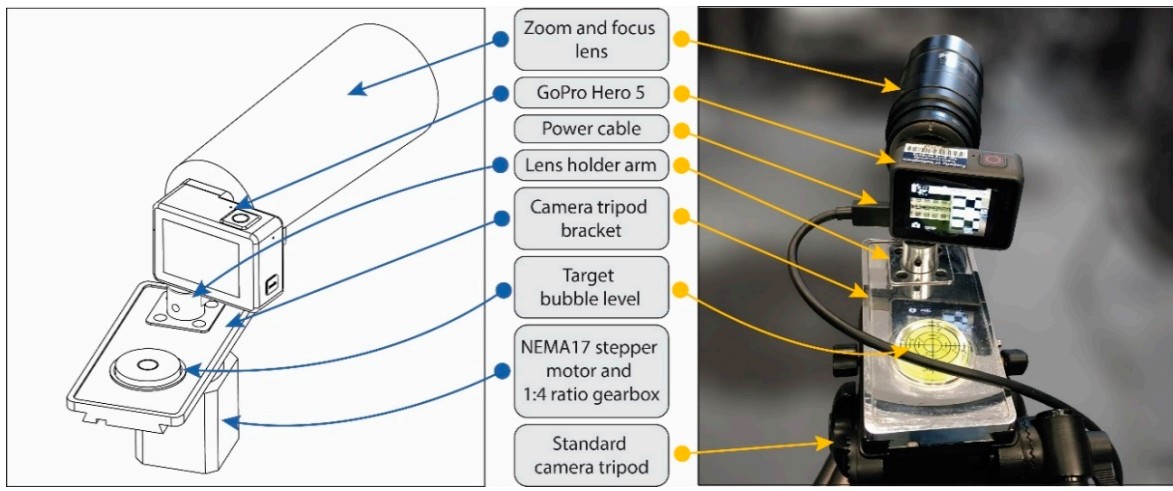

**Figure 2.** Proposed robotic camera system: design (**left**) and photo (**right**).

An encoder, which is typically used to run a closed-loop positioning system, is avoided to reduce the complexity and costs of the camera system. Instead, an open-loop system is used, and the positional calibration is done manually by aligning the robotic mount in the horizontal plane using a target bubble level and capturing a single sequence of test images. The sequence of test images are inspected for correct focus, image misalignments and inclusion of the desired parts of the structure.

An automatic zoom lens is avoided to reduce costs. Camera zoom and focus are performed by the user physically rotating the settings on the zoom lens. After each physical change to the zoom lens settings, a sample image is taken, and the quality of the captured image is assessed by the user. If the image is well-focused and the region of interest is in the frame, then calibration is considered complete. If the image does not center the target region of interest and is not in focus, the user performs fine tuning of the manual settings of the zoom lens, and the calibration process is repeated.

Software control is done using Python scripting, which follows the flowchart with pseudo-steps in Figure 3. Image sequences and camera rotations are set by the user. They can be either finite (via time limit or total image sequence/camera rotation limits) or infinite (the system continues capturing image sequences until a manual user break). The GoPro Application Programming Interface (API) for Python is an open-source software module that enables the connection and control of a GoPro camera via a Wi-Fi connection. During the setup, if the camera or robotic mount is not ready, the setup process is terminated, and the user receives an error message requesting manual corrections of the faults. At a no-fault scenario, the RCS proceeds with capturing image sequences. The rotation angles for each image sequence are predetermined by the user in advance of the experiment and are coded into the Python script. The camera rotation is visually judged by the user so that each new region of structure is in the frame of each new image.

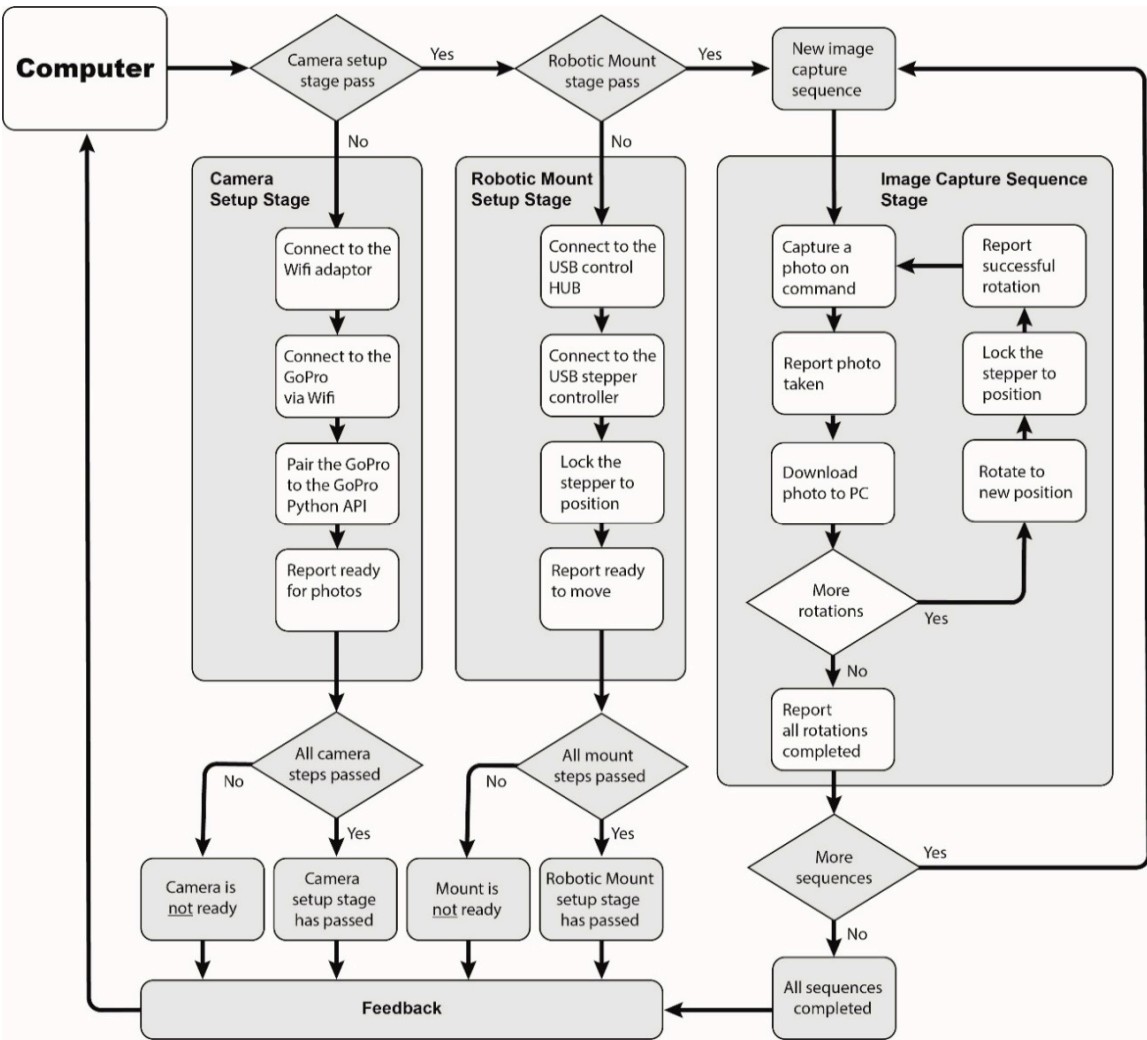

**Figure 3.** Robotic camera system (RCS) flowchart with pseudo-steps.

At the start of each new image capture sequence, a new folder is created. Each new photo is saved in the respective image sequence folder and given a unique name. Each stage of the image capture sequence is reported to the user, including the names and locations of captured images. Continuous file name reporting allows for an on-going manual inspection of images during the camera operation as the user can load each file to inspect it during the image capture sequence. Once all required image capture sequences are complete, the user is informed, the stepper motor is powered down, and the camera is set to stand-by.

## 2.2. Data Processing

Data processing steps for deriving the structural response from the images collected with the RCS are shown in Figure 4. A data folder for each image sequence or assumed time step (*i*), which can last as long as required to capture the entire structure or a part of it, is created. Collected images are sorted following the camera rotation (*J*), and an image list is created. The first image sequence is assumed to represent the baseline conditions of the structure, in which (a) regions of interest (ROI) and targets are identified, (b) targets are characterized, (c) and image homography for each *J* is computed.

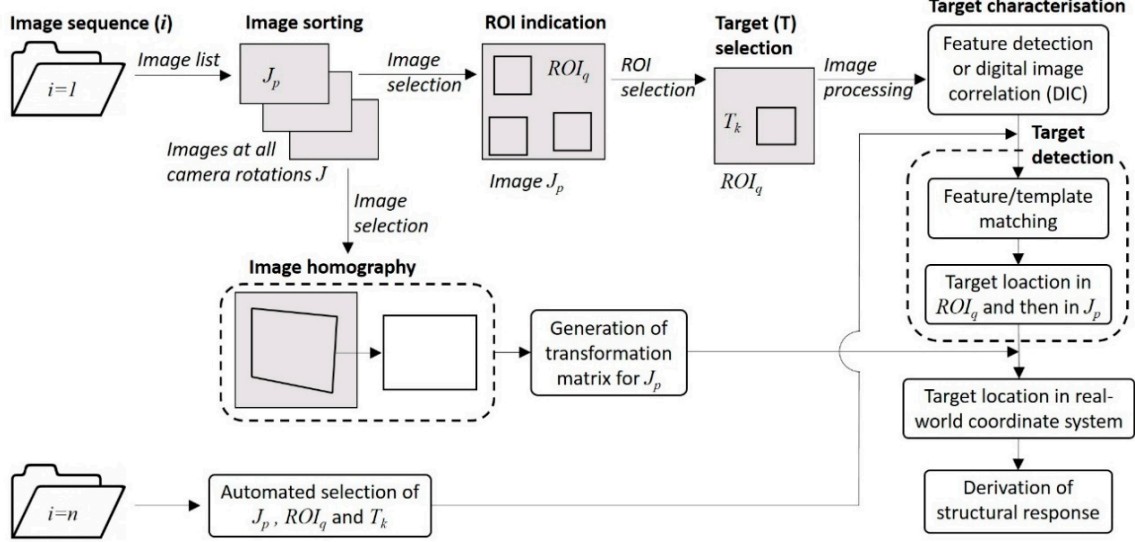

**Figure 4.** Data processing flowchart

The most important step is an accurate characterization of targets and their detection in consequent images. It is initiate with an automatic or manual identification of ROIs in $J_p$ images (where *p* is the rotation sequence number). ROIs are created to reduce computational costs, i.e., only ROI from each camera rotation is load instead of the entire image when detecting the corresponding target. The location of a target within $ROI_q$ (where *q* is the number of ROI in $J_p$ image) is defined. Both (a) feature detection algorithms such as mini-eigenvalues to detect mathematical features in an object of interest or target (b) and digital image correlation or template matching technique, in which the target of interest is located in a ROI, can be considered for characterizing targets. The target location is found from either the arithmetic averages of mathematical feature or the center location of the template in the ROI. The principles of both object tracking algorithms are well known, and references can be found in [12,16]. The target location is transformed from the ROI coordinate system to a global/image coordinate system. ROIs and targets for each $J_p$ are stored in the memory for analysis of consequent image capture sequences.

In the image homography phase coordinates of at least four widely distributed reference points, which, for example, can be bridge joints obtained from structural plans, are needed. Locations of the reference points are selected in the image frame. Coordinates of the locations of the reference points in the image and known coordinates of the same points obtained from the plans are used to generate a transformation matrix. The matrix is then used to convert locations of targets from a pixel coordinate system to a real-world coordinates system.

In new image sequences each selected $J_p$ image undergoes an automatic target detection. The stored targets for corresponding ROIs and Js are sought, their locations are detected and converted to the real-world coordinate system using the predefined transformation matrix for the corresponding $J_p$. From each image sequence, a response measurement is extracted. Consecutively obtained response

measurements form structural response time-histories, which can be used to analyze the performance of the structure.

## 3. Experimental Study Results

The measurement accuracy of the proposed robotic camera system is evaluated on a laboratory structure, which is equipped with a contact sensing system. The structure, the contact sensor system, and camera systems are introduced. The structural response measured with contact sensors and calculated from images collected with a smartphone and the RCS are compared and discussed.

### 3.1. Laboratory Setup

A simply supported timber beam with the length, width, and height of 1000 mm, 20 mm, and 40 mm, respectively, serves as a testbed for the performance evaluation of the proposed RCS. The beam has rectangular laser engravings mimicking structural targets such as bolts in steel bridges. Engravings are four 3 mm × 3 mm squares with a 14 mm offset in both horizontal and vertical directions. The load is applied manually placing weights at the mid-span of the beam. The following load steps are considered: 0 N, 50 N, 75 N, 85 N, 90 N, 85 N, 75 N, 50 N, and 0 N. Structural response is collected at 1 Hz with five linear variable differential transformers or displacement sensors (denoted as $D_i$, where $i = 1, 2, \ldots, 5$) and each load step with three foil strain gauges ($SG_i$, where $i = 1, 2, 3$). Figure 5 is a sketch of the beam with the contact sensing system.

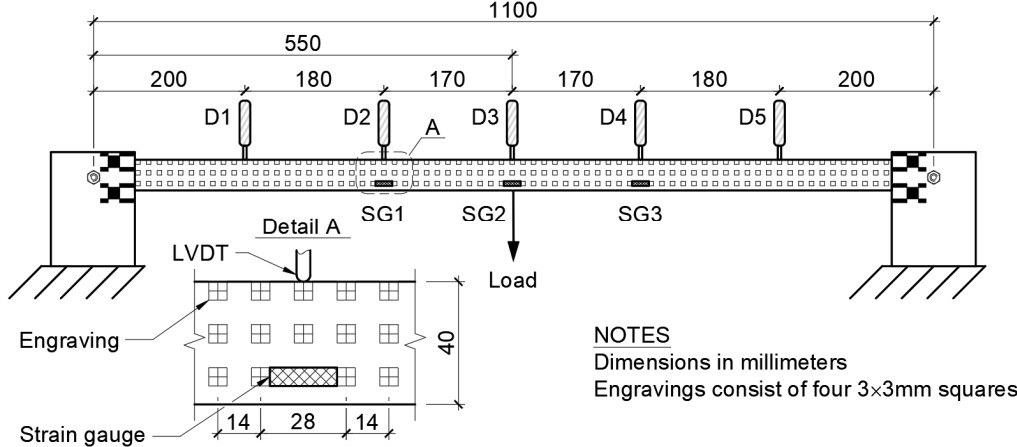

**Figure 5.** A sketch of the test beam with its principal dimensions and sensor locations.

Figure 6 shows the set-up of the vision-based measurement collection system. It consists of a Samsung S9+ smartphone and the proposed RCS. The smartphone is set 1 m away from the center of the beam with its field of view capturing the entire beam. For static experiments, when loads are applied stepwise, images can be taken at low frequencies, hence, significantly reducing data size and its processing time. Image frames usually provide more pixels than video frames. For example, the selected smartphone can record videos at 4k (3840 × 2160 pixels) and image of 4032 × 2268 pixels. Therefore, the smartphone is set to capture still images at 0.2 Hz. The RCS is set at a 3 m distance from the center of the beam. The zoom lens is set to 135 mm. The RCS carries out nine image collection sequences – one sequence per load step. In total, 20 image frames are captured per image sequence. The resolution of RCS images is 4000 × 3000 pixels.

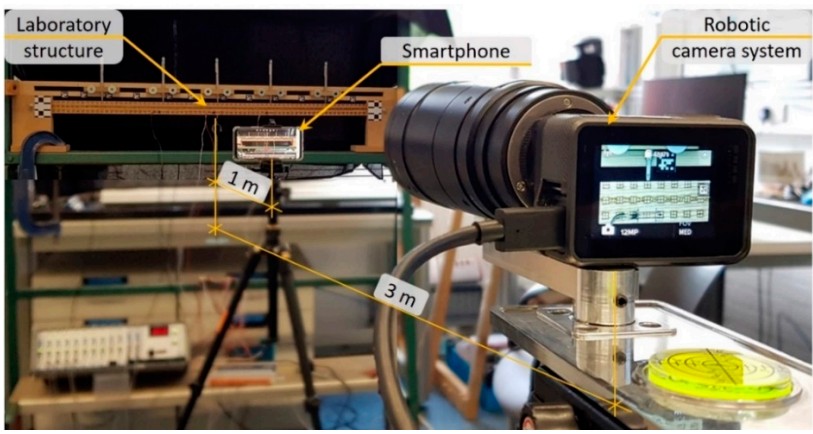

**Figure 6.** Experimental setup.

Stationary targets, consisting of Aruco codes (see Figure 7a) attached to the frame holding the displacement sensors, are placed next to each displacement sensor and at intervals no larger than 100 mm. The purpose of stationary targets is (a) to evaluate if the RCS rotations are accurate (b) and remove measurement errors from displacement calculations.

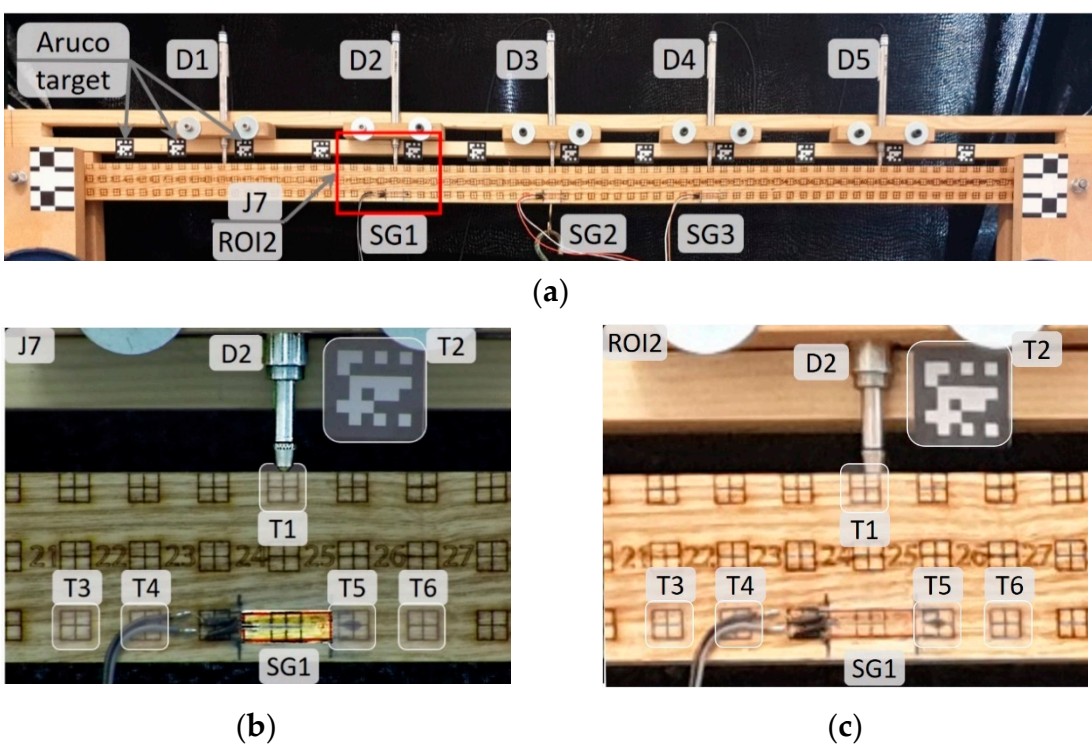

**Figure 7.** Annotated images collected with vision-based monitoring systems. (**a**) A cropped image captured with a smartphone. Sensor names are provided next to sensor locations. The red rectangle shows the part of the beam captured at J7 and in ROI2. (**b**) and (**c**) targets and sensor locations in J7 and ROI2, respectively.

*3.2. Data Processing*

The data processing strategy introduced in Section 2.2 is adapted. An image frame, which includes the beam and its supports, captured with a smartphone, is shown in Figure 7a. Five smartphone ROIs and camera rotations, one at each contact sensor location, are selected to analyses and compare measurement accuracies. The images that are taken with the RCS cover approximately the same areas

as selected ROIs in smartphone images. ROI2 serves as a demonstrator of a typical ROI selected from smartphone images. Five camera rotations (J4, J7, J10, J13, and J16) are considered in the comparison study. These camera rotations (similarly to smartphone ROIs) contain six targets, of which five are engravings, and one is a stationary target (see Figure 7b,c). In Table 1, smartphone ROI and RCS J numbers are listed together with contact sensor names and numbers, which are included in the image frame or region of interest. For clarity, only the region of J7 and ROI2 is drawn in Figure 7a.

**Table 1.** Contact sensors in RCS rotations and smartphone ROIs.

| Camera rotation ($J_p$) | J4 | J7 | J10 | J13 | J16 |
|---|---|---|---|---|---|
| Smartphone $ROI_q$ | ROI1 | ROI2 | ROI3 | ROI4 | ROI5 |
| Sensor in $J_p$ and $ROI_q$ | D1 | D2, SG1 | D2, SG2 | D4, SG3 | D5 |

RCS rotation error ($E_c$) on the x-axis (horizontal) and y-axis (vertical) for two camera rotations J4 and J7 and target T2 (denoted as J4T2 and J7T2) near displacement sensor locations for each load step/image sequence are shown in Figure 8a. Vertical errors increase at each load step. Horizontal errors do not follow a similar trend to horizontal errors. Additionally, the magnitude of horizontal errors is slightly smaller than that of vertical errors. Mean RCS rotation error ($E_{c,mean}$) is calculated using Equation (1):

$$E_{c,mean} = \frac{\sum_{i=1}^{n-1}|l_{xy,i} - l_{xy,i+1}|}{n-1} \tag{1}$$

where $l_{xy,i}$ is the location of the target on x- or y-axis at $i^{th}$ image sequence and $n$ is the total number of image sequences. Horizontal and vertical $E_{c,mean}$ values for each load step are given in Figure 8b. The smallest $E_{c,mean}$ is found for J7, which, however, has no particular explanation. Overall the mean RCS rotation error is approximately one pixel.

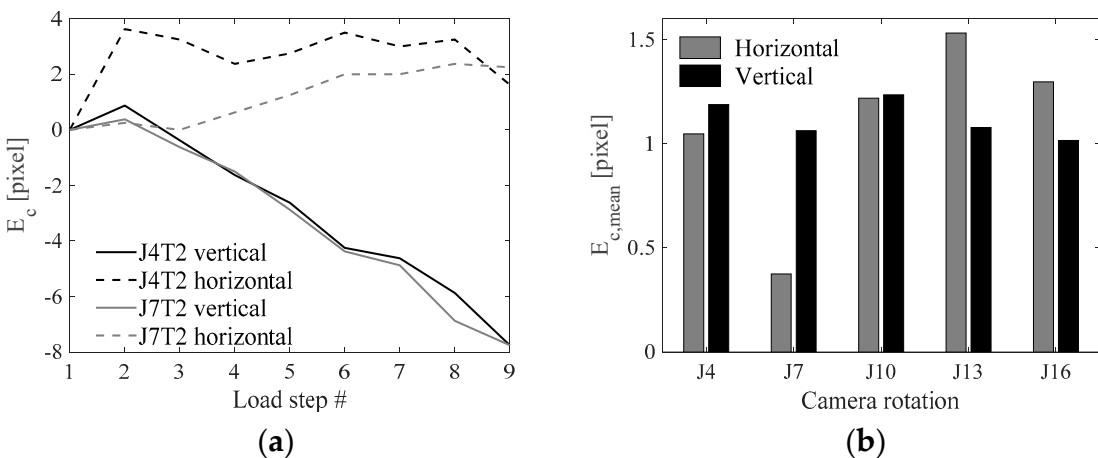

**Figure 8.** RCS rotation error. (**a**) Vertical and horizontal at J7T2. (**b**) Mean vertical and horizontal camera rotation errors for five rotations of the camera (see Table 1).

Coordinates of each target within a selected ROI are converted to the image coordinate system. A projective transformation approach is selected for mapping target locations on the image coordinate system to the real-world coordinate system. This step requires the provision of known coordinate points and their corresponding real-world measurements. Once the geometric transformation is applied to target location/coordinates, their displacements can be read in real-world units.

Strain ($\varepsilon$) or the ratio of a change of the length over the original length between two targets is a parameter which is expected to remain immune to camera rotation errors. Strain at $i^{th}$ load step for target combination ($t$) consisting of targets $Tk$ and $Tm$ is calculated using Equation (2), in which the

distance (*d*) between two targets is derived from their *x* and *y* locations on the image coordinate system. Structural strains are small and expressed in parts per million (µε). Therefore, an average of strains for multiple targets located as far from each other as possible, but within boundaries of an image frame, is taken as a representative strain value. For examples, an average value of strains between targets T3 and T5, T3 and T6, T4 and T5, and T4 and T6 is said to represent strain at corresponding strain gauge locations.

$$\varepsilon_{i,t} = \frac{d_{i,t} - d_{i-1,t}}{d_{0,t}} \tag{2}$$

$$d_{i,t} = \sqrt{(Tk_i(x) - Tm_i(x))^2 + (Tk_i(y) - Tm_i(y))^2} \tag{3}$$

### 3.3. Structural Response

The structural response is obtained for all targets. However, only a few targets are used to compare measurement accuracy between contact sensors. Figure 9a shows a plot of vertical displacements measured with displacement sensor D2 and computed from target T1 at RCS rotation J4 and smartphone ROI2. The plot shows both time steps and load steps. The displacements during load application and removal are slightly different. This is due to the nature of the beam reaction to applied/remove loads. When the load is removed the beam does not return to its original shape. The vertical displacement is 0.05 mm. Although vertical displacements with both camera systems are in a good agreement with the displacement sensor data, the RCS offers higher measurement accuracies than the smartphone.

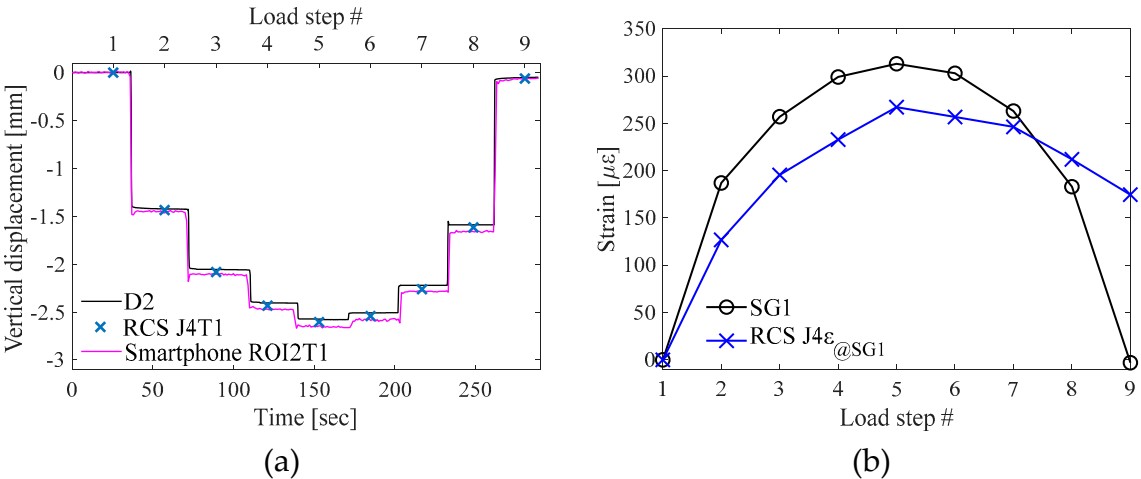

**Figure 9.** (**a**) Vertical displacements at D2 location. (**b**) Strain time-histories.

Average strain values from the camera rotation J4 (J4$\varepsilon_{@SG1}$) is plotted together with the strain gauge SG1 measurements. Strains calculated from target displacements in smartphone images are erroneous and do not have a clear load pattern. Therefore, they are excluded from the strain comparison graph. Overall, strains computed from RCS images closely follow the loading pattern and accurately show changes in response even at 10 N and 5 N loads. 10 µε drop from load step 5 to 6 is accurately captured with the RCS. An exception is the last image sequence at which the RCS has a very high measurement error.

Root-mean-square errors (RMSE) between contact sensors and both the smartphone and the RCS are calculated for target T1 located in the vicinity of a contact sensor (see Table 1). The measurement error ($E_m$) is derived using the range of measured response (*r*) for the selected contact sensor and average RMSE between the contact sensor and the corresponding target (see Equation (4)):

$$E_m = \frac{1}{n} \frac{\sum_{i=1}^{n} RMSE_i}{r} \tag{4}$$

Figure 10 shows displacement measurement errors for the RCS and the smartphone. Overall, displacement measurements are computed very accurately. Results demonstrate that the overall average $E_m$ of the smartphone (3.3%) is at least twice as that of the RCS (1.4%). As seen from Figure 9b strains are not very accurate. Measurement errors for strains collected with SG1, SG2 and SG3, and the RCS at rotations J7, J10, and J13 are 56 $\mu\varepsilon$, 119 $\mu\varepsilon$, and 69 $\mu\varepsilon$, respectively, which are equal to 18%, 24%, and 29% of the strain range at the corresponding sensor location. Figure 9b shows that the most significant difference is in the last step. The same phenomenon, which significantly affects the overall measurement error values, is observed for the other two sensor locations.

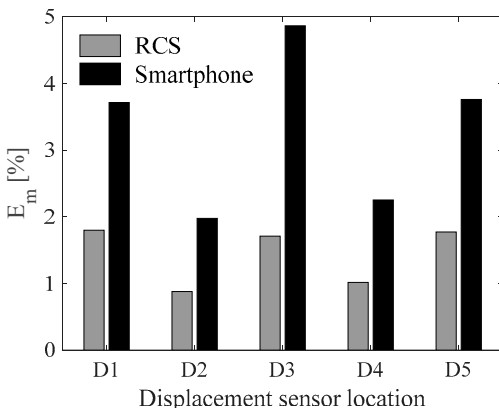

**Figure 10.** The measurement error ($E_m$) between displacement sensors and cameras.

## 4. Discussion

### 4.1. Data Processing Challenges and Achievements

The smartphone and RCS field of views are 77° and 2.37°, respectively. The wide lens affects the view angle of the image. For example, the inner sides of the supports holding the beam are discernible (see Figure 7a), hence, creating a fisheye effect which needs to be removed through camera calibration. The narrow-angle lens, in turn, provides a much more realistic view of parts of the structure than the smartphone camera. The number of pixels, which has a direct impact on the measurement accuracy, is much higher for images collected with the RCS in comparison to smartphone ROIs equivalent to the RCS image size. The difference in the image quality is noticeable when comparing J4 and ROI2 (see Figure 7b,c). An engraving, which is a 6 × 6 mm, consists of 210 × 210 pixels and 22 × 22 pixels in the RCS images and smartphone images, respectively.

The RCT rotation angle error ($E_a$) is found using the relationship between $E_c$, which is converted to the engineering units such as millimeters, and the camera distance to the target ($y$) (see Equation (5)). Considering the average $E_c$ being approximately 1 pixel or 0.031 mm, $E_a$ is 1/1000[th] degree or 10 $\mu$rad. The rotation angle error is very small; however, further testing is needed to find if the error persists in escalating after a large number of image capture sequences:

$$E_a = 2 \tan^{-1} \frac{E_c}{y} \tag{5}$$

The key challenge is to measure strains. In this experimental study, the maximum strain, 500 $\mu\varepsilon$, is at the mid-span of the beam when 90 N load is applied. The distance between T3 and T6 (see Figure 7b) is 70 mm or 2265 pixels, hence, 500 $\mu\varepsilon$ are equal to $(2265 \times 500)/(1 \times 10^6) = 1.13$ pixels. Proprietary software such as the "Video Gauge" developed by Imetrum Ltd. and hardware can achieve a maximum of 1/500[th]-pixel resolution accuracy at perfect environmental conditions [23]. In this study, small load steps are distinguishable in strain measurements (see Figure 9b). The smallest change (from the load step 5 to the load step 6) is 10.4 $\mu\varepsilon$ or 1/50[th] pixel. Although it is ten times lower than the measurement

accuracy claimed by proprietary software, the open-source algorithms employed offer reasonably high precision being suitable for low-cost systems.

The accuracy of measurements collected with contact sensors depends on the quality of the installation, sensors, and data acquisition system. It is important to recognize the possibility of measurement uncertainties/errors in contact sensors when comparing the accuracy of measurements collected with contact sensors and computed from images. When assessing the long-term performance of a bridge under structural health monitoring, measurement-histories or signals, which are treated for outliers and with moving average filters, are preferred. Signal trends are important when assessing measurements for anomaly events, which could indicate accelerated fatigue of or damages to a bridge. In such scenarios, the measurement accuracy of individual time points is not as important as the signal trend. The RCS is therefore recommended for the collection of long-term temperature-driven structural response.

### 4.2. A Vision of an Enhanced Three-Axis Robotic Gimbal

The proposed RCS demonstrates that accurate measurements can be collected reliably at relatively low costs. The total cost of the RCS prototype is £1050, out of which the modified GoPro camera costs £520, the zoom lens costs £320, the USB control system costs £78, the gearbox and stepper motor cost £32, and the aluminum alloy mounting bracket and camera arm manufactured in-house at an estimated cost of £100.

More expensive robotic camera systems could include a high-performance control system, precision stepper motors, high-quality anti-backlash gearboxes, anti-backlash electromechanical brakes, and high-quality precision encoders. These types of enhancements would produce a robotic system with improved positional accuracy.

A three-axis robotic camera gimbal could offer the possibility to capture more data (e.g., multiple rows of image captures from a single position) than a single axis system developed in this study. Three-axis control would also remove positional errors in the y-axis seen with the current RCS. An enhanced RCS would feature three rotation axes while also adding robotic zoom and focus controls. The central position of the camera sensor is at the center of all axes. An action camera and zoom/focus lens would be enhanced with robotic zoom and focus mechanisms offering remote control of these features. Stepper motors and gearboxes with the addition of encoders to provide closed-loop feedback of rotations would be used for each axis. A central three-axis precision tilt sensor would check and record the final camera positions before capturing the consecutive image sequence. Electromechanical brakes would be added to remove the continuous load from the gearboxes and would provide stronger hold when locked in a position. In the event of a power cut, the brakes would also secure the gimbal in place (an important safety feature as the system weight is increased). Finally slewing bearings would be added at each joint to reduce wear on the gearboxes and to aid smooth rotation throughout the entire range of motion. Such enhancements would marginally increase system costs. The highest costs would be for manufacturing the camera mount. The envisioned three-axis robotic gimbal is shown in Figure 11.

### 4.3. Future Research

The proposed RCS is at its embryonic stage. It has much room for further improvements and fine tunings. The control system can be developed as suggested in Section 4.2. The system can be enhanced with artificial intelligence (AI), which is a combination of situational awareness and creative problem solving [24]. It can, therefore, be considered that the more situationally aware a system is, the better it will be able to perform. The awareness of an element of infrastructure such as a bridge could enable a RCS to focus on specific, predefined tasks. For example, one of RCS tasks is to capture real-time deformations and surface cracks of a joint of a bridge during truck crossings. Information of an approaching truck is passed by another camera, which is a constituent of the bridge monitoring system. Other task could be monitoring long-term structural response at selected joints.

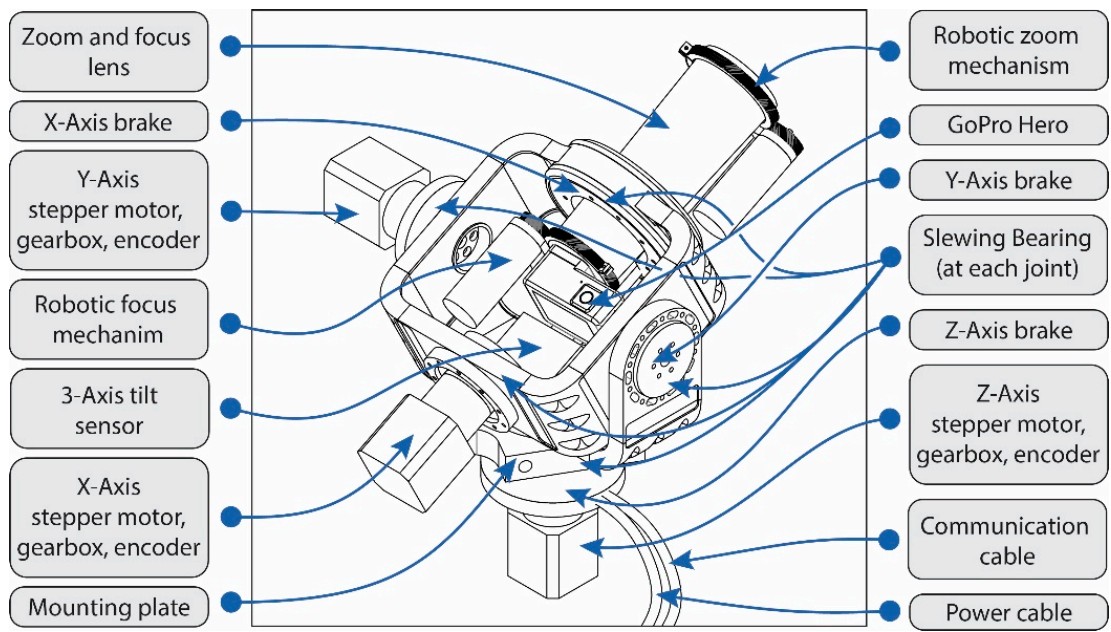

**Figure 11.** A vision of three-axis robotic gimbal.

Future work will build on a network of low-cost camera monitoring systems with the development of a prototype AI, which is capable to assess autonomously the performance of civil structures. Multi-RCSs could be employed for the 3D reconstruction of (i) objects and surfaces, (ii) and crack propagation [25,26] for condition assessment of structures. Other benefits of a low-cost- nationwide system of robotic cameras and AI monitoring could include traffic control support, traffic accident reporting, and crime prevention. Additionally, any access permissions that could be gained from existing cameras already in place (e.g., speed check cameras, security cameras, social media uploads) could be inputs given to an AI.

## 5. Conclusions

The paper introduces a robotic camera system (RCS) for accurate structural response collection in structural health monitoring. The RCS is composed of a modified GoPro with a zoom lens, a robotic mount, and open-source software controls. The data processing strategy for the analysis of RCS captured images is presented and discussed. The performance of the RCS is evaluated on a laboratory timber beam. The beam response is monitored with contact sensors and a smartphone. The main conclusions drawn from this study are as follows:

- The proposed RCS is designed, manufactured, and assembled using low-cost parts. It is controlled using opens source scripts, has repeatable positioning, can capture good quality experimental data, and is simple to use. The RCS has a slight rotation error of $1/1000^{th}$ degree.
- The low-cost RCS provides very accurate vertical displacements. The overall measurement error of the RCS is 1.4%, which is more than two times smaller than that of the smartphone. In this study, 1.4% corresponds to 0.03 mm.
- Strains can be computed from images collected with the RCS; however, their accuracy is not as high as that of displacements. The loading pattern is clearly discernible in the strain measurements. The smallest strain step is found to be 10 $\mu\varepsilon$. The proposed RCS has a very good potential for applications in long-term measurement collection.

**Author Contributions:** All sections and experiments R.K. and C.F.

**Funding:** This study was supported by the School of Architecture, Design and the Built Environment, Nottingham Trent University, and no external funding was received.

**Acknowledgments:** The authors would like to express their gratitude to Jordan Fewell for manufacturing the camera mount plate and lens holder arm.

**Conflicts of Interest:** The authors declare no conflict of interest.

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
