# Peer review of "A Low-Cost Robotic Camera System for Accurate Collection of Structural Response"

_inventions, doi:10.3390/inventions4030047_

Round 1

Reviewer 1 Report

In response to the limitations of bridge monitoring equipment, the authors propose a low-cost robotic camera system for accurate measurement collection of structural response. The system includes hardware structure, software system and data processing strategy, which includes data sorting, image processing and response generation. That is a good idea.

The robot camera system is low in cost, simple in control, easy to use, and has good precision. The description in the manuscript is adequate. I suggest receiving it after minor revision. The problems are as follows:

1. Line 53 – “Usually in 53 bridge monitoring, only a small part of the structure is considered providing very localized response 54 measurements.” Please give proof of relevant references.

2. Line 103 – “and the positional calibration is done manual” “manual” should be changed to “manually”. And how did the positional calibration done?

3. Line 106 – “and camera zoom and focus is done manually.” How did camera zoom and focus done?

4. Line 120 – “Once all required image capture sequences have been completed the user is informed, the stepper motor is powered down, and the camera is set to stand-by” How is the rotation position determined? Do you need to set different angles for different objects in advance?

5. Line 134 to line 136. How did the ROI created? According to the manuscript, the ROI was determined first, and then the target was detected in the ROI. If so, is the ROI artificially given?

6. Line 140 – “The target location is found either from the arithmetic averages of mathematical feature or the center location of the template in the ROI.” Why? Please describe it.

7. Line 145 – “The coordinates can be obtained from structural plans. Locations of the reference points are selected in the image frame.” How to choose these points?

8. Line 204 – “(b) and (a) targets and sensor locations in J7 and ROI2, respectively” “(a)” should be changed to “(c)”.

9. In Fig. 7, according to the position of the frame drawn by the author, should the J7 box in Fig. 7(a) be changed to J6? Now the figure looks jumbled, text comments please point to the corresponding part with a line. Please indicate the location of each ROI in the image.

10. I think the author should quote these two papers:

[1]. Chen, M., Tang, Y., Zou, X., Huang, K., Li, L., & He, Y.. High-accuracy multi-camera reconstruction enhanced by adaptive point cloud correction algorithm. Optics and Lasers in Engineering. 2019; 122: 170–183.

[2]. Tang Y-C, Li L-J, Wang C-L, Chen M-Y, Feng W-X, Zou X-J, Huang K-Y. Real-time detection of surface deformation and strain in recycled aggregate concrete-filled steel tubular columns via four-ocular vision. Robotics and Computer-Integrated Manufacturing. 2019; 59: 36-46.

Reviewer 2 Report

The paper discusses a low-cost robotic camera system (RCS) for accurate
measurement collection of structural response. The paper is well written with appropriate sections and discussions. Reviewer has few minor comments,

Line 5 of the abstract is confusing and needs to be rephrased.

The paper has few grammatical errors. Please do one round of English proof reading and eliminate the errors.

Major comment: Reviewer feels that the novelty of the proposed method is limited. Please add some additional new options or functionalities that will generate interest among readers.

Future work to be specified

Round 2

Reviewer 2 Report

Authors have incorporated all the suggestions.